# Hyaluronic Acid-Coated Nanoliposomes as Delivery Systems for Fisetin: Stability, Membrane Fluidity, and Bioavailability

**DOI:** 10.3390/foods13152406

**Published:** 2024-07-29

**Authors:** Yan Sun, Xinghui Shen, Jiaqi Yang, Chen Tan

**Affiliations:** Key Laboratory of Geriatric Nutrition and Health (Beijing Technology and Business University), Ministry of Education, China-Canada Joint Lab of Food Nutrition and Health (Beijing), School of Food and Health, Beijing Technology and Business University (BTBU), Beijing 100048, China; 18630277818@163.com (Y.S.); shenxinghui2022@163.com (X.S.); listen.y3@outlook.com (J.Y.)

**Keywords:** liposomes, hyaluronic acid, fisetin, delivery system, bioaccessibility

## Abstract

Fisetin has shown numerous health benefits, whereas its food application is constrained by water insolubility, poor stability, and low bioaccessibility. This work investigated the potential of hyaluronic acid (HA)-coated nanoliposomes for the encapsulation and delivery of fisetin. It was observed that HA can adsorb onto the liposomal membrane through hydrogen bonding and maintain the spherical shape of nanoliposomes. Fluorescence analysis suggested that the HA coating restricted the motion and freedom of phospholipid molecules in the headgroup region and reduced the interior micropolarity of the nanoliposomes but did not affect the fluidity of the hydrophobic core. These effects were more pronounced for the HA with a low molecular weight (35 kDa) and moderate concentration (0.4%). The HA coating improved the storage and thermal stability of the nanoliposomes, as well as the digestive stability and bioaccessibility of the encapsulated fisetin. These findings could guide the development of HA-coated nanoliposomes for the controlled delivery of hydrophobic bioactives such as fisetin in functional foods.

## 1. Introduction

Fisetin (3,3′,4′,7-Tetrahydroxyflavone) is a biologically active flavanol found in a variety of fruits, vegetables, nuts, and teas [1]. Numerous scientific studies have shown its promising efficacy in anti-cancer, anti-angiogenesis, anti-oxidation, anti-aging, and immune suppression applications [2]. For example, it can enter the bloodstream after oral administration to reduce the risk of various cancers by regulating oxidative stress and inflammatory genes [3]. It is also used as a dietary supplement at a dose of 20 mg/kg/day to provide neuroprotection [4]. Despite these beneficial effects, the application of fisetin in the food industry is limited by its water insolubility, poor stability, and low bioaccessibility [5].

A liposome is an ideal delivery system that can encapsulate hydrophilic and lipophilic compounds in different parts of the phospholipid bilayer [6,7]. The bioactive compounds encapsulated in liposomes can be protected from external environmental stressors and released to a specific location [8,9]. It has been shown that fisetin can be located in the phospholipid bilayer, and its biological effects are significantly improved after liposomal encapsulation [10]. However, conventional liposomes face disadvantages toward aggregation and fusion during storage and in the gastrointestinal environment, which negatively affect the preservation and release of bioactives.

The combination of liposomes and biopolymers is an effective strategy to increase the delivery performance of liposomes [11]. Hyaluronic acid (HA) is a naturally derived linear glycosaminoglycan composed of N-acetylglucosamine and D-glucuronic acid with β-1,4-inter-glycoside bonds [12]. As an endogenous substance, HA has low toxicity, non-immunogenicity, biocompatibility, and biodegradability [13]. In 2021, HA was approved as a food and drink additive in China [14]. The oral administration of HA can help maintain collagen production [15], promote skin and eye moisturization [16], and relieve dry mouth [17]. HA also has the ability to interact with the lipid head groups and induce the interdigitation of the liposomal membrane [18,19]. Moreover, HA can bind to the extracellular domain of the cluster determinant, conferring the HA-coated liposomes with good mucoadhesive properties [20,21,22].

In this study, we aimed to stabilize nanoliposomes through a HA coating and increase the bioaccessibility of loaded fisetin. We prepared HA-coated nanoliposomes for the encapsulation and delivery of fisetin. The effects of HA’s molecular weight (MW) and concentration on the liposomal membrane fluidity and polarity were investigated using fluorescence probes, including 1,6-diphenyl-1,3,5-hexatriene (DPH), 1-anilinonaphthalene-8-sulfonate (ANS), and pyrene. Next, the storage stability, release, and digestive behavior of nanoliposomes were evaluated. The results demonstrated that the HA coating improved the delivery performance of nanoliposomes for fisetin, which could broaden the applications of fisetin in functional foods.

## 2. Materials and Methods

### 2.1. Materials

Lecithin from pure egg yolk (purity 98%) was purchased from Macklin (Shanghai, China). Fisetin (purity > 96%) was provided by Aladdin industrial corporation. Fluorescence probes, including ANS, pyrene, and DPH, were purchased from Sigma Aldrich. HA of different MWs (3, 35, 90–100, 150–250, and 1000–1500 kDa) was provided by Shanghai Yuan Ye Bio-Technology Co., Ltd. (Shanghai, China). A dialysis bag with a 500 kDa molecular weight cut-off was purchased from MYM Biological Technology Company. Methanol and acetic acid were of HPLC grade (Beijing MREDA Technology Co., Ltd., Beijing, China). Tween 80, ethanol, and other reagents were of analytical grade.

### 2.2. Preparation of Fisetin-Loaded Nanoliposomes 

The fisetin-loaded nanoliposomes (F-NLs) were prepared by the thin-film evaporation method [23]. Briefly, lecithin, Tween 80, and fisetin were dissolved in 2 mL of ethanol. The ethanol was then evaporated through a rotary evaporator at 40 °C to obtain a thin film. To completely remove the solvent, the film was further dried in an oven at 30 °C. Afterward, 10 mL of ultrapure water was added to hydrate the film, and the pH was adjusted to 7.0. Finally, the F-NLs were produced by an ultrasonic instrument (UW 100, Sonics & Materials, Bandelin, Germany) with a sonotrode (TS 113, Bandelin, Berlin, Germany) in an ice bath for 2 min at 40 W/cm^2^. The final mass ratio of lecithin and fisetin was 25:1, and the concentration of fisetin was 0.8 mg/mL.

To prepare the HA-coated fisetin-loaded nanoliposomes (F-HA-NLs), the HA powder was dissolved in a phosphate buffer of pH 7.0 at different concentrations (0.1–1.5%). The HA solution was then mixed with F-NLs at a volume ratio of 1:10 and stirred overnight. All of the samples were stored at 4 °C before use.

### 2.3. Physical Characterization

The average particle size, polydispersity index (PDI), and zeta potential were measured by dynamic light scattering (DLS) using a Malvern NanoZS90 zeta-sizer equipped with a He/Ne laser. The liposomal samples were diluted 120 times with phosphate buffer to avoid particle aggregation. Each sample was measured three times.

Fourier-transform infrared (FTIR) spectra were obtained using the single-reflection attenuated total reflectance (ATR) crystal (Nicolet iS10, Thermo Fisher, Waltham, MA, USA). The spectra were scanned from 4000 to 400 cm^−1^ with a resolution of 8 cm^−1^.

The morphology of the nanoliposomes was characterized by a transmission electron microscope (TEM). A drop of the diluted sample was placed on a copper screen covered with a supporting membrane. The observation was carried out using an electron microscope (Japan Electronics Co. Ltd., Tokyo, Japan) with an 80 kV acceleration voltage.

### 2.4. Liposomal Membrane Fluidity

A fluorescence technique was used to measure the fluidity of the liposomal membranes. Referring to a previous method [24], stock solutions of ANS, DPH, and pyrene were prepared with concentrations of 6 × 10^−3^, 1 × 10^−4^, and 4.7 × 10^−2^ mol/L, respectively. The nanoliposomes were homogeneously mixed with the probe solutions. The final concentrations of each probe were 3 × 10^−4^ mol/L (ANS), 1 × 10^−5^ mol/L (DPH), and 2.4 × 10^−3^ mol/L (pyrene), respectively. The mixtures were kept at 37 °C for 30 min. It is worth noting that the ANS and pyrene solutions needed to be dried by nitrogen before mixing with the nanoliposomes to avoid the damaging effects of ethanol on the liposome membrane. Fluorescence anisotropy was determined using a fluorescence spectrometer (Hitachi F-7000, Tokyo, Japan) equipped with a polarization filter. The excitation wavelengths (E_x_) and emission wavelengths (E_m_) were set at the following conditions: E_x_ = 337 nm and E_m_ = 480 nm for ANS and E_x_ = 358 nm and E_m_ = 425 nm for DPH. The slit width was 5 nm. Various polarizing parts, including I_0,0_, I_0,90_, I_90,90_, and I_90,0_, were measured separately. The fluorescence anisotropy (r) of DPH and ANS were calculated according to the following equations:(1)$$r=\left(I_{0,0}-{\mathrm{GI}}_{0,90}\right)/\left(I_{0,0}+2{\mathrm{GI}}_{0,90}\right)\;G=I_{90,90}/I_{90,0}$$

For pyrene, the E_x_ was 375 nm, and the scanning range of E_m_ was 400–700 nm. The ratio I_1_/I_3_ was calculated by dividing the fluorescence intensity near the first emission peak, I_1_ (373 nm), by the third emission peak, I_3_ (383 nm). This ratio indicates the polarity of the environment in which the pyrene is exposed.

### 2.5. Encapsulation Efficiency

The free fisetin was determined by using an extraction method [25]. The liposomal sample was mixed with n-pentane in a centrifuge tube and vortexed for 2 min. Then, the mixture was centrifuged at 6000× *g* at 25 °C for 5 min. The upper layer of n-pentane was collected. This operation was repeated three times, and the pentane solutions were then combined. The content of fisetin was determined by high-performance liquid chromatography (HPLC-1260, Agilent, Santa Clara, CA, USA) with methanol and acetic acid (2% *v/v*) at a volume ratio of 85:15 as the mobile phase, a flow rate of 0.5 mL/min, and a wavelength of 363 nm. The encapsulation efficiency (EE) of fisetin was calculated using the following equation:(2)$$\mathrm{EE}\;\left(\%\right)=\left(\mathrm{total}\;\mathrm{amount}\;\mathrm{of}\;\mathrm{fisetin}-\mathrm{free}\;\mathrm{fisetin}\right)/\mathrm{total}\;\mathrm{amount}\;\mathrm{of}\;\mathrm{fisetin}\times 100$$

### 2.6. Stability Assays

To evaluate the stability of nanoliposomes, the following conditions were applied: (i) storage at 4 °C and 37 °C for 15 days in the dark, and (ii) heating at 90 °C for 20 min [26]. The residual amounts of fisetin in the treated samples were determined by HPLC. The retention rate (RR) of fisetin was calculated as follows:(3)$$\mathrm{RR}\;\left(\%\right)=\;\mathrm{encapsulated}\;\mathrm{fisetin}\;\mathrm{after}\;\mathrm{storage}/\mathrm{encapsulated}\;\mathrm{fisetin}\;\mathrm{initially}\;\mathrm{prepared}\times 100$$

### 2.7. Lipid Peroxidation Inhibition Ability

Lipid peroxidation inhibition capacity was determined by measuring thiobarbituric acid reactive substances (TBARSs). A solution composed of thiobarbituric acid (15%, *w/v*), hydrochloric acid (1.8%, *v/v*), and trichloroacetic acid (0.37%, *w/v*) was added to 1 mL of the liposomal sample. The mixture was heated at 100 °C for 30 min, rapidly cooled in an ice bath, centrifuged at 6000 g at 4 °C for 10 min, and filtered. The absorbance of the filtrate was recorded at 535 nm by a UV spectrophotometer (UV-2600i, SHIMADZU Co. Ltd., Kyoto, Japan). A_c_ and A_s_ are the absorbances of blank nanoliposomes and fisetin-loaded nanoliposomes, respectively.
(4)$$\mathrm{TBARS}\;\mathrm{inhibition}\;\left(\%\right)=\left(A_{c}-A_{s}\right)/A_{c}\times 100$$

### 2.8. In Vitro Release 

The release behavior of nanoliposomes was evaluated by a dialysis method [27]. First, the dialysis bag was kept in the aqueous solution overnight before use in order to ensure complete wetting of the bag. Next, 2 mL of the liposomal sample was placed in the dialysis bag, transferred to 8 mL of phosphate buffer of pH 1.3, and incubated for 3 h. Then, the dialysis bag was transferred to a phosphate buffer of pH 7.4 and incubated for 7 h. At specific intervals, 1 mL of dialysis solution was taken out of the bag and replaced with an equal volume of buffer. The content of fisetin in the dialysis solution was determined by HPLC [28]. The release rate of fisetin was calculated as follows:(5)$$\mathrm{Release}\;\mathrm{rate}\;\left(\%\right)=\;\mathrm{released}\;\mathrm{fisetin}/\mathrm{total}\;\mathrm{fisetin}\times 100$$

### 2.9. Simulated Digestion Experiment

The simulated gastric fluid (SGF) and simulated intestinal fluid (SIF) were prepared based on a previous study [29]. The SGF was composed of sodium chloride (2 g/L), hydrochloric acid (0.008 g/L), and pepsin (3.2 g/L). The pH of the prepared solution was adjusted to 1.3 using hydrochloric acid. The SIF was composed of sodium hydroxide (1.81 g/L), pancreatin (4.76 g/L), bile salts (5.16 g/L), and potassium dihydrogen phosphate (8.09 g/L). First, 5 mL of the liposomal sample was mixed with 15 mL of SGF for 1 h. Afterward, the pH was adjusted to 7.5 using NaOH to simulate intestinal fluid digestion. The pH-stat method was used to keep the pH value of the system at 7.5 by continuously adding 0.1 mol/L NaOH solution during the 2 h of intestinal digestion. The consumed amount of NaOH was recorded as the digestion time progressed. The amount of free fatty acids (FFAs) was calculated based on the consumed volume of NaOH:(6)$$\mathrm{FFAs}\;\mathrm{released}\;\left(\%\right)={\;V}_{\mathrm{NaOH}}\left(t\right)C_{\mathrm{NaOH}}M_{w,\mathrm{lipid}}/2m_{\mathrm{lipid}}\times 100$$
where m_lipid_ is the total mass of phosphatidylcholine (g), M_w,lipid_ is the mean molecular weight of phosphatidylcholine (g/mol), C_NaOH_ is the concentration of NaOH solution used in the titration (mol/L), and V_NaOH_ (t) is the volume of NaOH solution consumed at digestion time (mL).

Next, the digesta was centrifuged at 10,000× *g* at 4 °C for 30 min. The digesta was divided into two parts after centrifugation, the upper layer being a transparent micelle layer containing fisetin and the lower layer being a dense insoluble substance containing undigested samples, free fatty acids, and bile salts. The stability and bioaccessibility of fisetin were determined by the following equations:(7)$$\mathrm{Stability}\;\left(\%\right)=\;\mathrm{amount}\;\mathrm{of}\;\mathrm{fisetin}\;\mathrm{in}\;\mathrm{digestive}\;\mathrm{fluid}/\mathrm{total}\;\mathrm{amount}\;\mathrm{of}\;\mathrm{lipid}\;\times 100$$
(8)$$\;\mathrm{Bioaccessibility}\;\left(\%\right)=\mathrm{amount}\;\mathrm{of}\;\mathrm{fisetin}\;\mathrm{in}\;\mathrm{the}\;\mathrm{micellar}\;\mathrm{layer}/\mathrm{total}\;\mathrm{fisetin}\;\times 100$$

### 2.10. Statistical Analysis

All tests were performed three times, and the results were expressed as mean and standard deviations. Statistical significance (*p* < 0.05) was determined using a one-way ANOVA using SPSS software (Version 17.0).

## 3. Results

### 3.1. Physical Properties 

The morphological features of the nanoliposomes were visualized by TEM (Figure 1). The F-NLs exhibited a rectangular shape with an average particle size of 80 nm. The presence of HA slightly increased the particle size of the nanoliposomes. The HA of high molecular weight typically consisted of longer polysaccharide chains, resulting in an increase in hydrodynamic particle size when forming complexes with liposomes [30]. At an MW of 1000–1500 kDa, the particle size was the largest. The F-HA-NLs maintained good dispersion, and the surface was rough with a thin coated layer. The particle size and size distribution determined by the DLS technique (Appendix A) were similar to the observations from TEM. The PDI values reflecting the particle size distribution were around 0.3 for all formulations, indicating good dispersion [31].

Fisetin exhibited intrinsic fluorescence at 540 nm when bound to liposome membranes [32]. Figure 2a shows that the intrinsic fluorescence intensity of fisetin did not significantly change in the presence of HA at 3 kDa, while the fluorescence intensity progressively decreased with the increasing MW of HA. The reason for this could be that the HA coating on the liposomal surface partially screened the intrinsic fluorescence of fisetin. This finding was consistent with previous work, where the combination of HA and berberine–oleanolic acid induced a shift in the absorption peak and the formation of a complex system [33].

The fluorescence anisotropy, r, was inversely related to the liposomal membrane fluidity [34]. The larger the r value, the more stable the liposome was. Here, we used DPH and ANS to evaluate the fluidity of the hydrophobic and surface regions of the liposomal membrane, respectively (Figure 2b). DPH exhibited high lipophilicity and was localized in the hydrophobic domains of amphiphilic molecules [35]. The r value of DPH was almost unchanged for all the tested MWs of HA, indicating that the incorporation of HA had no effect on the internal environment and structure of the liposomal membrane. However, the r value of ANS showed a trend of increasing first and then decreasing. At a MW of 35 kDa, the r value of ANS was the largest, suggesting that the HA coating restrained the movement of phospholipid molecules in the polar head. The data from fluorescence anisotropy indicated that HA caused a change in the surrounding environment of the liposome membrane, which may be related to the interaction of HA with the liposomal membrane. In summary, the modifying effect of HA was stronger on the liposomal surface than on the hydrophobic core. In a previous study, the coating of HA on the surface of liposomes improved their transdermal and targeted delivery capabilities for undecylenoyl phenylalanine [36]. Figure 2c shows the I_1_/I_3_ ratio deduced from the pyrene spectra. A higher ratio indicates a higher polarity in the pyrene environment [37]. The I_1_/I_3_ ratio increased with increasing MW from 3 to 35 kDa, whereas the ratio decreased beyond this MW range. We hypothesized that a HA of 35 kDa could form a dense network and a highly hydrophobic domain inside the nanoliposomes. This effect may help to inhibit the penetration of the polar pro-oxidant molecules into the liposomal membrane and protect the encapsulated fisetin [38,39]. A previous study found that the hydrophilic coating of HA on the surface of liposomes can reduce the bilayer fluidity and membrane permeability, thus prolonging the release of paclitaxel from liposomes [40].

Figure 2d shows that the encapsulation efficiency of F-HA-NLs were maintained at a high level with values of 90–95% at 3, 35, and 90–100 kDa. When the MW of HA was 150–250 kDa and 1000–1500 kDa, the encapsulation efficiency decreased to 79% and 74%, respectively. This result is consistent with the change in the I_1_/I_3_ ratio. A possible explanation is that the HA with a high MW increased the polarity of the liposomal membrane and consequently decreased its retaining ability to the hydrophobic fisetin. It is also possible that the HA with a high MW changed the conformation of phospholipids, loosened the arrangement of phospholipid molecules, and caused a leakage of fisetin. It was also observed that the loading capacity of gallic acid was increased after HA was coated on the chitosan nanoparticles [41]. HA can also improve the ligand structure and create more space to accommodate more curcumin in cancer cells [42].

The FTIR spectra of various formulations are presented in Figure 2e. The characteristic peak at 1096 cm^−1^ was related to the C-O-C band associated with the -PO_4_ group of the phospholipid molecule [43]. When the MW was above 35 kDa, the intensity of the C-O-C band became higher, indicating that the HA with a high MW disturbed the arrangement of phospholipid molecules and exposed more -PO_4_ groups. The peak at 1611 cm^−1^ was associated with the vibration of the C=C band on the aromatic ring of fisetin. After HA coating, this peak shifted to 1615 cm^−1^, which is probably due to the interaction between the -C=O group in fisetin and HA [44]. The peak from -OH and -NH stretching in nanoliposomes shifted from 3392 to 3390 cm^−1^ after the addition of HA, which suggests the formation of hydrogen bonding between the liposomal membrane and HA [45]. This result is consistent with the finding that hyaluronic acid interacted primarily with phospholipid head groups through hydrogen bonding [46]. Similar spectral changes were reported in the HA-coated liposomal nanoparticles [47]. The zeta potential data also show that the surface charges of liposomes were almost unchanged at different HA concentrations, implying that hydrogen bonding was the main driving force (Appendix A).

We also evaluated the effect of HA concentration on the properties of nanoliposomes. Different concentrations of HA have different characteristics [48]. All samples had good dispersibility and a negative surface charge at the tested concentrations (Appendix A). With the increase in HA concentration, the particle size of the nanoliposomes decreased and then increased. At a HA concentration of 0.4%, the particle size was the smallest. Moreover, the intrinsic fluorescence intensity of fisetin decreased with increasing HA concentration, suggesting stronger interactions between HA and fisetin (Figure 3a). At high concentrations, more HA covered the surface of the liposomes and weakened the fluorescence intensity of fisetin. Note that the anisotropy, r, values of DPH did not change with HA concentration, which further proves that DPH had no effect on the internal structure of the liposomal membrane. For ANS, its anisotropy, r, value was the largest when the HA concentration was 0.4%, implying that the fluidity of the liposomal surface was restricted at this concentration (Figure 3b). A previous study showed that coating with HA led to an increase in fluorescence anisotropy and enhanced the stability of cholesterol-contained liposomes [49]. The I_1_/I_3_ ratio increased with increasing HA concentration up to 0.4% and then leveled off. Similarly, the presence of HA decreased the polarity around pyrene in the niosomes, which may be due to the decrease in micro-viscosity [50]. In other words, the nanoliposomes had the lowest interior polarity at 0.4% (Figure 3c). Correspondingly, the encapsulation efficiency of fisetin was highest at 4%, with a value of 92% (Figure 3d).

### 3.2. Stability

Liposomes are unstable systems that are easy to degrade, aggregate, and fuse during thermal processing or after oral administration, resulting in the leakage of encapsulated bioactives. In this regard, we determined the retention rate of fisetin during storage for 15 days at different temperatures in the dark (Figure 4a). At 4 °C, the retention rates of F-NLs and F-HA-NLs were 94% and 92% after 15 days, respectively. At 37 °C and 90 °C, the retention rates of F-HA-NLs were significantly higher (*p* < 0.05) than those of F-NLs. An earlier study also evaluated the stabilizing effect of HA on quercetin-loaded polylactic-co-glycolic acid nanoparticles. It was found that HA provided spatial stability for the nanoparticles during storage at 4 °C and 30 °C for 3 months [51]. It is known that thermal treatment can accelerate the motion of phospholipid molecules and cause the leakage of fisetin. However, the coating of HA provided a physical barrier for liposomal vesicles and reduced their aggregation during heating. The hydrogen bonding between HA and the liposomal membranes also slowed down the motion of the phospholipid molecules and helped maintain their structural integrity.

Phospholipids are easily subjected to oxidative degradation by the external environment, generating malondialdehyde and subsequently TBARS [52]. Figure 4b shows that the presence of HA enhanced the oxidative stability of nanoliposomes. The TBARS inhibition rates of F-NLs and F-HA-NLs were 52.4% and 62.7%, respectively. This is attributed to the fact that the dense covering from the HA layer hindered the interactions of external pro-oxidant compounds with the lipid–water interface. In addition, the low polarity in the liposomal membrane resulting from the HA coating could have also prevented the penetration of oxygen into the liposomes [53].

### 3.3. In Vitro Release

It is essential to control the release of bioactive compounds for high adsorption in the intestine. Figure 5 shows that an initial burst release occurred within 1 h at pH 1.3 due to the release of unencapsulated and surface-related fisetin. Afterward, fisetin was slightly released from both the F-NLs and F-HA-NLs. This is understandable because a low gastric pH is considered to exert little effect on the integrity of liposomes [54]. At pH 7.4, fisetin exhibited a slow and sustained release from the F-HA-NLs. After incubation for 9 h, the release rates were 99% for F-NLs and 94% for F-HA-NLs. The slower release of F-HA-NLs may have been induced by the limited bilayer mobility, which was similar to the results of targeting delivery using HA-modified liposomes [55]. It has also been proven that the paclitaxel-loaded liposomes had sustained release characteristics after being modified by HA [56].

### 3.4. In Vitro Digestion

Correlating in vitro-simulated release with in vitro-simulated digestion can provide a more complete understanding of the release behavior of bioactive compounds in a simulated in vivo environment. The oral absorption of fisetin depends on the proportion of fisetin in the digestive juice and micelle phase in the small intestine [57]. Thus, we used the pH-state method to monitor FFA release from nanoliposomes during the process of intestinal digestion. It was observed that the FFA in all formulations was released rapidly at an initial digestion of 30 min (Figure 6a). The released amounts of FFA from the nanoliposomes were much higher than those from bulk oil, suggesting that the nanoliposomes increased the extent and rate of lipid hydrolysis [58]. Additionally, the liposomal encapsulation increased the chemical stability of fisetin during digestion (Figure 6b). The stability values of fisetin in bulk oil, F-NLs, and F-HA-NLs were 25%, 49%, and 52%, respectively. The better stabilizing effect of F-HA-NLs could be due to the fact that the coated layer of HA reduced the permeability of the lipid bilayer and improved the oxidative stability of fisetin during digestion. HA may also provide viscosity in the gastrointestinal tract and decrease the leakage of fisetin and subsequent exposure to the external environment.

Finally, we measured the bioaccessibility of fisetin by determining its concentration in the micelle phase of digesta. The bioaccessibilities of fisetin in bulk oil, F-NLs, and F-HA-NLs were 7.2%, 88.9%, and 92.5%, respectively (Figure 6c). Generally, there is a positive correlation between the degree of lipolysis and bioaccessibility [59]. More FFA release and a larger surface area of nanoliposomes increased lipase’s access to the liposomal membrane surface and the subsequent transfer of fisetin to the micelle phase. Additionally, the HA coating retarded the degradation of fisetin, allowing more fisetin release into the micelle phase and therefore more bioaccessibility.

## 4. Conclusions

This work investigated the effect of HA coating on the properties of fisetin-loaded nanoliposomes, which are dependent on the molecular weight and concentration of HA. After coating with HA of a MW of 35 kDa at a concentration of 0.4%, both the liposomal membrane fluidity and interior polarity decreased, which helped to compact the liposomal structure and reduce the penetration of polar pro-oxidant compounds. Thus, the HA coating conferred the nanoliposomes with slow release, strong protection for fisetin during in vitro digestion, and high bioaccessibility. These findings suggest that HA of moderate molecular weight is a promising coating material for nanoliposomes. The biological function of the encapsulated fisetin will be investigated in the future through cellular and animal experiments.

## Figures and Tables

**Figure 1 foods-13-02406-f001:**
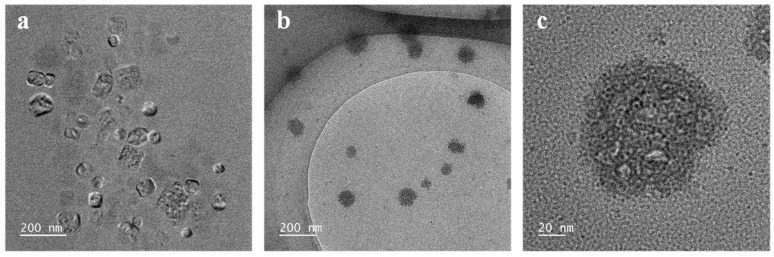
TEM images of (**a**) F-NLs and (**b**,**c**) F-HA-NLs.

**Figure 2 foods-13-02406-f002:**
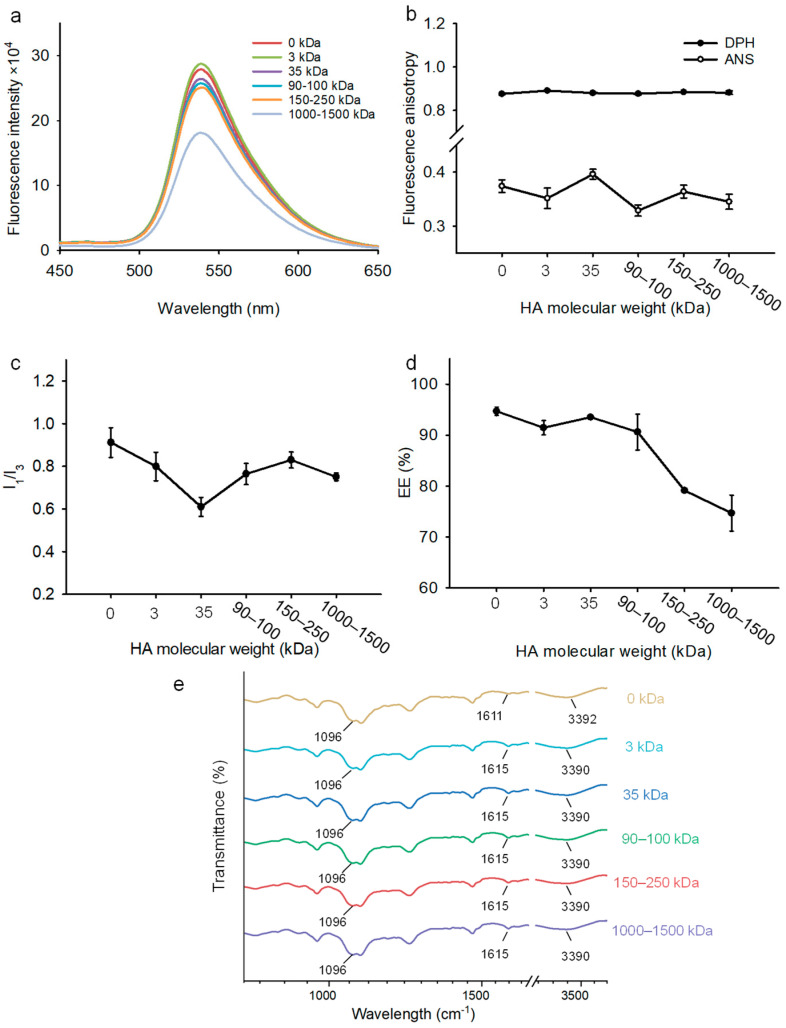
Effect of HA MW. (**a**) Intrinsic fluorescence spectra of fisetin. (**b**) Fluorescence anisotropy of DPH and ANS. (**c**) Polarity parameter, I_1_/I_3_, of pyrene. (**d**) Encapsulation efficiency of fisetin. (**e**) FTIR spectra. Each point represents the mean value ± standard deviation (n = 3).

**Figure 3 foods-13-02406-f003:**
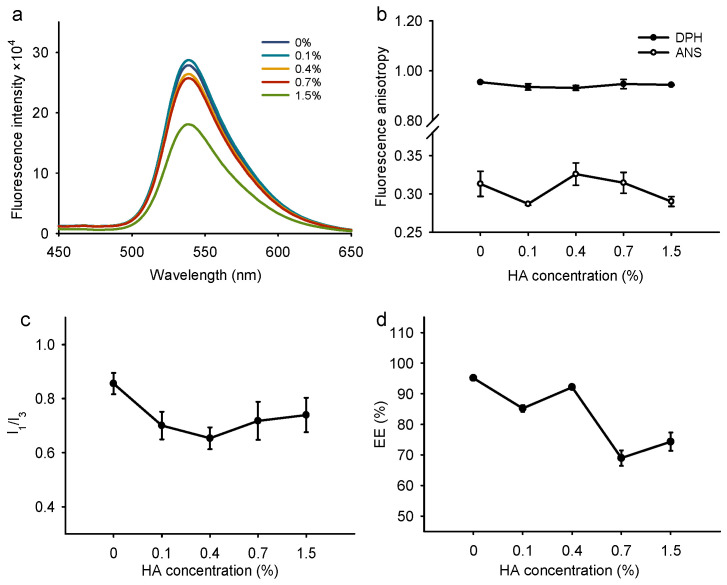
Effect of HA concentration. (**a**) Intrinsic fluorescence spectra of fisetin. (**b**) Fluorescence anisotropy of DPH and ANS. (**c**) Polarity parameter, I_1_/I_3_, of pyrene. (**d**) Encapsulation efficiency of fisetin. Each point represents the mean value ± standard deviation (n = 3).

**Figure 4 foods-13-02406-f004:**
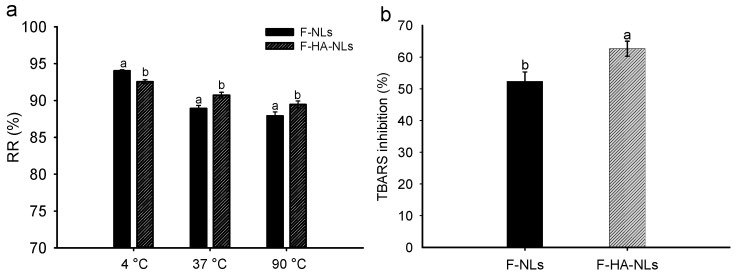
(**a**) Storage stability of F-NLs and F-HA-NLs at 4 °C, 37 °C, and 90 °C. (**b**) Lipid anti-oxidant capacity of the F-NLs and F-HA-NLs. At each temperature, the retention of fisetin in F-NLs and F-HA-NLs was compared. Different letters represent a significant difference (*p* < 0.05). Each point represents the mean value ± standard deviation (n = 3).

**Figure 5 foods-13-02406-f005:**
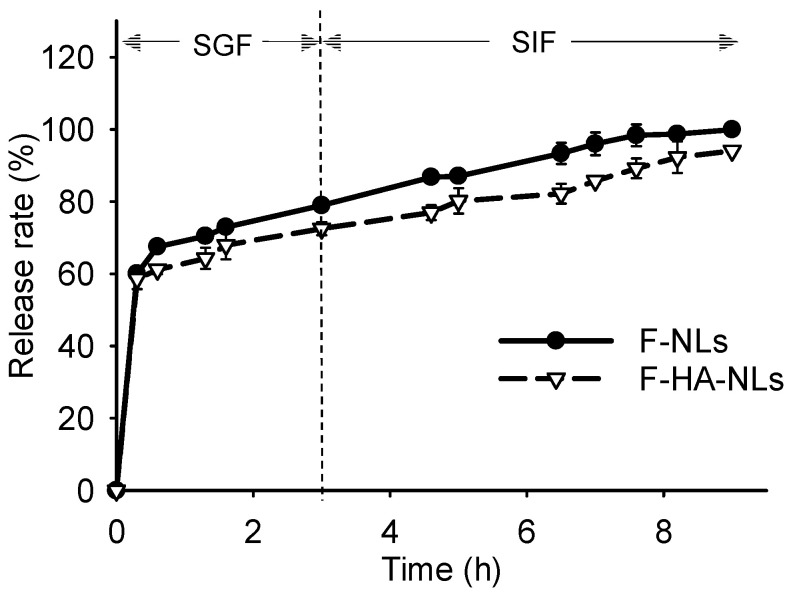
In vitro release profiles of F-NLs and F-HA-NLs at pH 1.3 and 7.4. Each point represents the mean value ± standard deviation (n = 3).

**Figure 6 foods-13-02406-f006:**
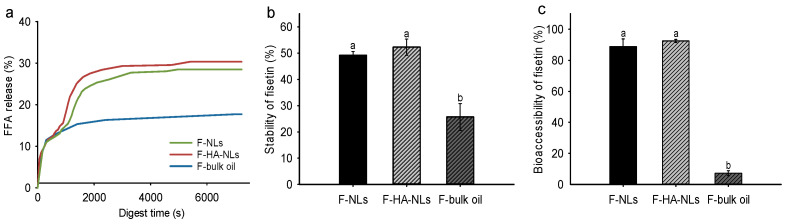
(**a**) Release profiles of FFA from bulk oil and nanoliposomes in the digestion process. (**b**) Stability and (**c**) bioaccessibility of fisetin in bulk oil and nanoliposomes after in vitro digestion. Different letters represent a significant difference (*p* < 0.05).

## Data Availability

The original contributions presented in the study are included in the article/Appendix A; further inquiries can be directed to the corresponding author.

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
