# Peer review of "Hyaluronic Acid-Coated Nanoliposomes as Delivery Systems for Fisetin: Stability, Membrane Fluidity, and Bioavailability"

_foods, 2024, doi:10.3390/foods13152406_

Round 1
Reviewer 1 Report
Comments and Suggestions for Authors
Abstract, 1st sentence: the use of “efficacy in numerous health benefits” is not appropriate and not reading well. Rewrite it.
Line 27: add “applications: after ‘immune suppression’
Lines 24-31: The discussion is all about Fisetin. Thus, from the 2nd sentence onwards, refer to “it” rather than reusing the name.
Line 51-57: instead of saying, we did this and that, rewrite the paragraph, with a research hypothesis, research objectives, and one sentence of obtained results and outcome.
Line 190: change to (r).
Figure 2e, the y-axis is missing.
Lines 294-296: A reference is not needed for a general statement. Thus, remove reference 47.
Author Response
Dear Reviewer:
Thank you for your letter and the comments on reviewers concerning our manuscript. Those comments are all valuable and extremely helpful for revising and improving our manuscript, and they also have an important guiding significance on our researches. We have studied your comments carefully and made corrections which we hope to be approved. Revised portions are marked by red highlight.
Abstract, 1st sentence: the use of “efficacy in numerous health benefits” is not appropriate and not reading well. Rewrite it.
Response: Thanks for suggestion, we have modified this sentence.
Line 27: add “applications: after ‘immune suppression’
Response: We have added it.
Lines 24-31: The discussion is all about Fisetin. Thus, from the 2nd sentence onwards, refer to “it” rather than reusing the name.
Response: Thanks for mention. We have used “it” to replace fisetin.
Line 51-57: instead of saying, we did this and that, rewrite the paragraph, with a research hypothesis, research objectives, and one sentence of obtained results and outcome.
Response: Thanks a lot for this professional suggestion, we have revised this paragraph based on the comments.
Line 190: change to (r).
Response: We have changed it.
Figure 2e, the y-axis is missing.
Response: The absorbance of FTIR spectra can not provide information for discussion, so we did not include it.
Lines 294-296: A reference is not needed for a general statement. Thus, remove reference 47.
Response: Thanks for mention. We have removed it.
Reviewer 2 Report
Comments and Suggestions for Authors
Results
Physics of properties
I) a broader discussion of the results with respect to other authors.
II ) Why the morphology is square for the case f (a-b) F-NLs and not (c-d) F-HA-NLs.
III) Homogenize the TEM scales all at 100 nm and/or 50 nm.
IV) In the results sections, there is no discussion with other authors of research already reported on similar materials.
a) Intrinsic fluorescence spectra of fisetin.
b) Fluorescence anisotropy
c) Polarity parameter
d) Encapsulation efficiency
e) FTIR spectra.
f) Stability
Author Response
- I) a broader discussion of the results with respect to other authors.
Response: Thanks a lot for this suggestion. We have added much discussion from other authors, which was copied in the question (IV).
II ) Why the morphology is square for the case f (a-b) F-NLs and not (c-d) F-HA-NLs.
Response: Thanks for this comment. We speculated that the coating of HA induced the shape change of liposomes, but no references were found. We will focus on studying the morphological evolution of liposome upon HA coating and relevant mechanisms in the near future.
III) Homogenize the TEM scales all at 100 nm and/or 50 nm.
Response: We have unified the scales of TEM images.
- IV) In the results sections, there is no discussion with other authors of research already reported on similar materials.
- a) Intrinsic fluorescence spectra of fisetin.
- b) Fluorescence anisotropy
- c) Polarity parameter
- d) Encapsulation efficiency
- e) FTIR spectra.
- f) Stability
Response: Thanks a lot for this suggestion. We have added more discussion according to the suggestion in the revised manuscript in red. Please also see the copied text here:
Line 191-193: This finding was consistent with previous work, where the combination of HA and berberine-oleanolic acid induced the shift of the absorption peak and the formation of the complex system[33].
Line 213-215: Previous study has found that the hydrophilic coating of HA on the surface of liposomes can reduce the bilayer fluidity and membrane permeability, thus prolonging the release of paclitaxel from liposomes[39].
Line 223-226: It has also been observed that the loading capacity of gallic acid was increased after HA coating on the chitosan nanoparticles[40]. HA can also improve the ligand structure and create more space to accommodate more curcumin in the cancer cell[41].
Line 238-239: Similar spectral changes were reported in the HA-decorated liposomal nanoparticles[46].